

# A combination of curcumin, vorinostat and silibinin reverses Aβ-induced nerve cell toxicity via activation of AKT-MDM2-p53 pathway

Jia Meng[1], Yan Li[2], Mingming Zhang[1], Wenjing Li[1], Lin Zhou[1], Qiujun Wang[1], Lin Lin[1], Lihong Jiang[1] and Wenliang Zhu[3]

[1] Department of General Medicine, Second Affiliated Hospital of Harbin Medical University, Harbin, China
[2] Department of Pharmacy, Fourth Affiliated Hospital of Harbin Medical University, Harbin, China
[3] Department of Pharmacy, Second Affiliated Hospital of Harbin Medical University, Harbin, China

Corresponding authors
Lihong Jiang,
jianglihong2006@163.com
Wenliang Zhu,
zhuwenliang@hrbmu.edu.cn

## ABSTRACT

Alzheimer's disease (AD) is a significant health issue for the elderly and becoming increasingly common as the global population ages. Although many efforts have been made to elucidate its pathology, there is still a lack of effective clinical anti-AD agents. Previous research has shown the neuroprotective properties of a combination of curcumin and vorinostat. In this study, nine other neuroprotective agents were investigated to examine whether a three-drug combination of curcumin, vorinostat, and a new drug is more advantageous than the previous two-drug combination in alleviating amyloid beta (Aβ)-induced nerve cell toxicity. Cell viability assay was performed to screen these agents, and further validation tests, including determination of cellular oxidative stress, apoptosis, and activity of the AKT/MDM2/p53 pathway, were performed. Among the nine candidate compounds, only silibinin at 1 μM reduced $A\beta_{25-35}$-induced toxicity in PC12 cells. The neuroprotective effects of 1 μM silibinin in combination with 5 μM curcumin and 0.5 μM vorinostat (CVS) was shown in PC12 cells, in which it decreased apoptosis and oxidative stress marker levels that were increased by 20 μM $A\beta_{25-35}$. Western blotting results showed that CVS pretreatment significantly increased the phosphorylation of AKT, BAD, and MDM2, which resulted in decreased intracellular expression of p53. Further, immunofluorescence results showed reduced p53 levels in the nuclei of PC12 cells following CVS pretreatment, indicating a reduction in the p53-mediated transcriptional activity associated with $A\beta_{25-35}$ exposure. In conclusion, our findings suggested that pretreatment with CVS protected PC12 cells from $A\beta_{25-35}$-induced toxicity through modulation of the AKT/MDM2/p53 pathway. Thus, CVS may present a new therapeutic option for treating AD.

# INTRODUCTION

Alzheimer's disease (AD) is a progressive neurodegenerative disorder characterized by extracellular deposition of amyloid beta (Aβ) plaques and intracellular aggregation of neurofibrillary tangles (*Sanabria-Castro, Alvarado-Echeverría & Monge-Bonilla, 2017*).
AD is the most common form of dementia among the elderly (*Mattson, 2004*), and is characterized by severe cognitive impairment and memory loss. AD is the fifth-leading cause of death in people over the age of 65 years, and it is estimated that 5.4 million people have AD in the United States (*Alzheimer's Association, 2016*). There are currently more than 7 million AD patients in China, and this number is growing (*Chan et al., 2013*; *Jia et al., 2014*). However, these issues are not specific to the United States and China because other countries are also facing problems associated with an aging population. The global growth of AD patients has contributed to a worldwide increase in nursing demand and economic burden (*Jia et al., 2018*). If no effective measures are taken, it is objectively estimated that one among every 85 individuals will suffer from AD by the middle of this century (*Brookmeyer et al., 2007*).

Despite the rapidly increasing prevalence of AD, there is a severe lack of therapeutic strategies to prevent AD or to reverse the development of Aβ plaques (*Agatonovic-Kustrin, Kettle & Morton, 2018*). To date, only three acetylcholinesterase inhibitors (donepezil, galantamine, and rivastigmine) and one noncompetitive N-methyl-D-aspartate receptor antagonist (memantine) have been approved by the US Food and Drug Administration (FDA) as treatments for AD (*Guzior et al., 2015*). Unfortunately, these approved drugs only elicit modest symptomatic improvement and temporary cognitive improvement in half of the patients with AD (*Blennow, De Leon & Zetterberg, 2006*). Given such limited options, a significant amount of research has focused on alternative therapies for AD. As a result, several natural compounds have been shown to exert promising neuroprotective effects mediated through the p53 pathway, making them promising candidate drugs for treating AD (*Jazvinšćak Jembrek et al., 2018*).

One such example is curcumin, a natural flavonoid isolated from Rhizoma Curcumae Longae, which exerts neuroprotective effects against Aβ-induced neurotoxicity in both cell and animal models (*Potter, 2013*). However, no clinical studies have shown the efficacy of oral curcumin in treating AD, which is likely due to the low bioavailability of curcumin in humans (*Ringman et al., 2012*). However, our research group has shown that curcumin and vorinostat, a histone deacetylase inhibitor, exert synergistic neuroprotective effects against Aβ toxicity in PC12 cells (*Meng et al., 2014*). The in vitro model of Amyloid beta-25-35 (Aβ$_{25-35}$)-induced PC12 cells used in the study is widely accepted and adopted by researchers in the field of neuroprotective agents (*Kim et al., 2015*; *Li et al., 2017*; *Zhao, Zhu & Guo, 2018*). Our previous study also suggested that co-administration of curcumin and vorinostat protects neural cells from Aβ-induced apoptosis by maintaining high phosphorylation of AKT serine/threonine kinase (AKT). It is noteworthy that the effect of the drug combination on the expression and transcriptional activity of the tumor protein p53 was not investigated. Given that simultaneous accumulation and cooperation of Aβ and p53 have been observed in patients with AD, it is likely that p53 plays a critical role in AD progression (*Ohyagi et al., 2015*). Furthermore, we suspect that through these synergistic mechanisms, the bioavailability of curcumin may be increased.

In the present study, the effects of curcumin were assessed in combination with other natural compounds to elucidate whether its neuroprotective effect is mediated through activation of the AKT/MDM2 proto-oncogene 2 (MDM2)/p53 pathway.

Furthermore, because many natural compounds such as resveratrol, piceatannol, genistein, quercetin, kaempferol, luteolin, apigenin, daidzein, and silibinin have been shown to have neuroprotective effects (*Wang et al., 2018*; *Fu et al., 2016*; *You et al., 2017*; *Ansari et al., 2009*; *Pate et al., 2017*; *Sawmiller et al., 2014*; *Zhao et al., 2013*; *Westmark, Westmark & Malter, 2013*; *Duan et al., 2015*), we explored whether the addition of one of these compounds would further enhance the neuronal benefit of curcumin and vorinostat.

## MATERIALS AND METHODS

### Materials

$A\beta_{25-35}$ peptide was purchased from Chinapeptides Corporation (Shanghai, China), diluted to a stock concentration of 1 mM in double-distilled $H_2O$, and incubated at 37 °C for 7 days to induce aggregation. Following incubation, stock solution of $A\beta_{25-35}$ was diluted in cell culture media to a final concentration of 20 $\mu$M, as described previously (*Jia et al., 2014*). Unless noted otherwise, all additional chemicals and reagents were purchased from Sigma Aldrich, Inc. (St. Louis, MO, USA).

### Cell culture and treatment

Rat pheochromocytoma (PC12) cells were grown on tissue culture-treated plates with Dulbecco's modified Eagle's medium/nutrient mixture F-12 (DMEM/F-12) supplemented with 7% fetal bovine serum (FBS) and 1% penicillin-streptomycin. The cells were maintained at 37 °C in an incubator with 5% $CO_2$. After reaching 80% confluence, the cells were cultured in serum-free medium for 12 h, and then pretreated for 1 h with either a single compound of interest or a combination of compounds. After the pretreatment, the cells were exposed to $A\beta_{25-35}$ peptide for 24 h.

### Cell viability assay

Cell viability was measured using the 3-[4,5-dimethylthiazol-3-yl]-2,5-diphenyltetrazolium bromide (MTT) assay. In brief, cells were seeded in a 96-well plate at a density of $1 \times 10^4$ cells/well and treated with compound(s) of interest, whereas the control cells were treated with an equal volume of DMEM-F12 with 7% FBS. Following a 24-h incubation, 10 $\mu$L of MTT (5 mg/mL) was added to each well, which was then incubated at 37 °C for 4 h. The medium was then discarded and replaced with 100 $\mu$L dimethyl sulfoxide (DMSO) to dissolve the crystals. Next, optical density was measured at 490 nm using a multi-mode microplate reader (SpectraMax M5; Molecular Devices Company, Sunnyvale, CA, United States).

### Determination of reactive oxygen species (ROS) levels

PC12 cells were pretreated with the following drug combinations for 1 h at 37 °C in DMEM: curcumin (5 $\mu$M) alone; a two-drug combination of curcumin (5 $\mu$M) and vorinostat (0.5 $\mu$M); or a three-drug combination of curcumin (5 $\mu$M), vorinostat (0.5 $\mu$M), and silibinin (1 $\mu$M). Following the pretreatment, $A\beta_{23-25}$ peptide was added to induce cell apoptosis, and intracellular ROS generation was assessed using a reactive oxygen species assay kit (Nanjing Jiancheng Institute of Biotechnology, Nanjing, China) according to the manufacturer's instructions. In brief, PC12 cells were washed twice with phosphate-buffered saline (PBS) and then incubated with 10 $\mu$M 2′,7′-dichlorodihydrofluorescein

diacetate (DCFH-DA) dye in serum-free DMEM-F12 for 1 h at 37 ° C. The cells were washed with PBS, and fluorescence was observed at 488 nm and recorded using confocal microscopy. Semi-quantification of ROS levels was performed by using the ImageJ software.

## Measurement of superoxide dismutase (SOD) and malondialdehyde (MDA) levels

Briefly, the treated PC12 cells were harvested and lysed in lysis buffer, and the resulting cell lysates were centrifuged at 3,000× g for 10 min at 4 °C. The resulting supernatant was collected for enzyme activity analysis. Levels of SOD and MDA were determined through the nitroblue tetrazolium assay using a total SOD Detection Kit (Beyotime Institute of Biotechnology, Shanghai, China) and MDA assay kit (Jiancheng Institute of Biotechnology, Nanjing, China), according to the manufacturers' instruction. Protein concentration was determined using a BCA Protein Assay Kit (Beyotime Institute of Biotechnology). One unit of SOD and MDA was defined as the amount of protein. SOD and MDA values were expressed as enzyme activity per milligram protein (U/mg).

## Western blotting assay

PC12 cells were washed with PBS, lysed with ice-cold radioimmunoprecipitation assay (RIPA) lysis buffer, and added to phenylmethanesulfonyl fluoride. The mixture was then centrifuged for 15 min at 14,000× g and 4 °C. Total cellular proteins in the supernatants were measured using a BCA protein assay kit (Beyotime Institute of Biotechnology) according to the manufacturer's instructions. Proteins were then separated using 10% sodium dodecyl sulfate-polyacrylamide gel electrophoresis, and protein bands were transferred to a polyvinylidene difluoride membrane (Millipore Corp., Bedford, MA, USA). Next, the membrane was blocked with 5% non-fat milk diluted in 1× tris-buffered saline with 0.1% Tween 1 h at room temperature (24 °C). The membranes were then incubated for 12 h at 4 °C with the following primary antibodies: rabbit polyclonal phospho-MDM2 antibody ($p$-MDM2; 1:1,000; Cell Signaling Technology, Danvers, MA, USA), mouse monoclonal p53 antibody (1:1,000; Cell signaling Technology), rabbit monoclonal phospho-AKT (ser473) antibody ($p$-AKT; 1:1,000; Cell Signaling Technology), rabbit monoclonal phospho-BAD (Ser112) antibody ($p$-BAD; 1:1,000; Cell Signaling Technology), and mouse monoclonal β-actin antibody (1:5,000; Abcam, Cambridge, MA, USA). The membranes were then washed and incubated at room temperature (24 °C) for 1 h with goat anti-rabbit IRDye® 800CW (1:10,000; Li-Cor Biosciences, Lincoln, NE, USA) or goat anti-mouse IRDye® 800CW (1:10,000; Li-Cor Biosciences). Proteins were visualized using the Odyssey Infrared Imager System (Li-Cor Biosciences), and the odyssey v1.2 software was used to quantify protein by measuring band intensity (area × optical density) in each group, with β-actin as an internal control.

## Transferase-mediated deoxyuridine triphosphate biotin nick-end labeling (TUNEL) assay

During apoptosis, cleavage of genomic DNA yields double-stranded DNA breaks that are identifiable by labeling the free 3′-OH termini with modified nucleotides in an enzymatic reaction; the TUNEL assay is based on this principle. In brief, PC12 cells were grown on

confocal dishes (NEST, Wuxi, China) in DMEM-F12 for 24 h, and then pretreated with the compounds of interest alone or in combinations for 1 h. The pretreated cells were then exposed to 20 μM Aβ$_{25-35}$ for 24 h at 37 °C and 5% CO$_2$. The cells were washed twice with PBS and then fixed with 4% paraformaldehyde (PFA) at room temperature (24 °C). The fixed cells were washed twice with PBS buffer, treated with 0.1% TritonX-100 to permeate cells, and then washed twice with PBS. Next, TUNEL reaction mixture (Beyotime Institute of Biotechnology) was added to the cells, which were then incubated for 60 min at 37 °C in a dark, humidified environment. Negative controls were prepared with equal volumes of labeling solution. After this incubation, the confocal dishes (NEST) were rinsed twice with PBS. Finally, the cells were analyzed using a confocal Laser Scanning Biological microscope FV1000 (Olympus, Tokyo, Japan) at an excitation wavelength of 488 nm.

## Immunofluorescence staining assay

Colocalization of p53 and MDM2 proteins was examined by immunofluorescence staining of the treated PC12 cells. In brief, PC12 cells were cultured on confocal dishes (NEST) at 37 °C for 24 h. After treatments with Aβ$_{25-35}$, curcumin (5 μM), vorinostat (0.5 μM), and silibinin (1 μM), as described previously, the cells were fixed with 4% PFA for 15 min at room temperature (24 °C), washed twice with PBS, and incubated with 0.1% Triton-X 100 for 15 min at room temperature (24 °C). The cells were then blocked with 1% bovine serum albumin in PBS for 30 min, and subsequently incubated with antibodies against p53 (1:1,000; Cell Signaling Technology) and MDM2 (1:400; Cell Signaling Technology) at 4 °C for 12 h with mild shaking. The cells were washed twice with PBS, then incubated with a secondary antibody at room temperature (24 °C) for 1 h; the secondary antibodies were donkey anti-rabbit 488 (1:1,000; Invitrogen, Camarillo, CA, USA) and donkey anti-mouse 594 (1:1,000; Invitrogen). To stain cell nuclei, the cells were incubated with 1 μg/ml 4′, 6-diamidino-2-phenylindole (DAPI) for 5 min. Next, the cells were washed twice with PBS and observed using a confocal Laser Scanning Biological microscope FV1000 (Olympus).

## Statistical analysis

All experiments were repeated five times, and all values are expressed as mean ± standard error of the mean (SEM). Statistical analysis was performed using GraphPad Prism v6.0 (GraphPad Software, Inc., San Diego, CA, USA). To assess differences among three or more groups, one-way analysis of variance (ANOVA) with Dunnett's test was used. Values of $p < 0.01$ were considered to indicate statistically significant differences.

# RESULTS

## Silibinin enhanced the neuroprotective effects of the combination of curcumin and vorinostat against Aβ$_{25-35}$-induced cytotoxicity

In the present study, nine natural compounds were investigated for synergistic activity with a drug combination of curcumin and vorinostat in Aβ$_{25-35}$-treated PC12 cells. The nine compounds, each had been documented to have neuroprotective effects (*Wang et al., 2018*; *Fu et al., 2016*; *You et al., 2017*; *Ansari et al., 2009*; *Pate et al., 2017*; *Sawmiller et al., 2014*; *Zhao et al., 2013*; *Westmark, Westmark & Malter, 2013*; *Duan et al., 2015*),

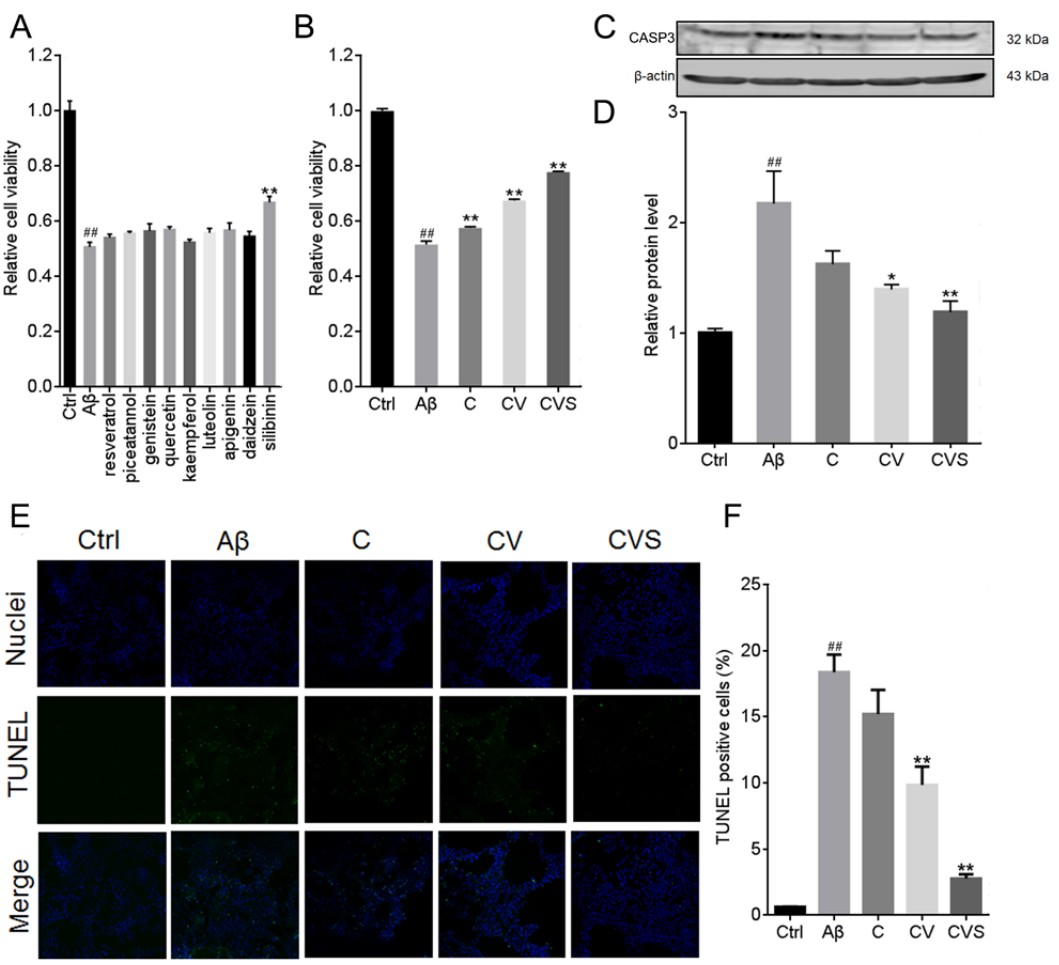

**Figure 1** **CVS treatment inhibits Aβ$_{25-35}$-induced PC12 cell apoptosis.** (A) Individual effects of 9 candidate compounds on cell viability of Aβ$_{25-35}$-treated PC12 cells. (B–F) Effects of CVS on cell viability (B), reversion of CASP3 expression (C, D), and apoptotic cell levels (E, F) in Aβ$_{25-35}$-treated PC12 cells. Each experiment was completed with a minimum of five replicates. Statistical significance is presented as; ## $p < 0.001$ versus Ctrl; *$p < 0.01$, **$p < 0.001$ versus Aβ. Abbreviations: Aβ, Aβ$_{25-35}$ treatment group; C, curcumin and Aβ$_{25-35}$ treatment group; CASP3, Caspase 3, Ctrl, control group; CV, curcumin, vorinostat and Aβ$_{25-35}$ treatment group; CVS, curcumin, vorinostat, silibinin and Aβ$_{25-35}$ treatment group.

included resveratrol, piceatannol, genistein, quercetin, kaempferol, luteolin, apigenin, daidzein, and silibinin. Our results indicated that at 1 μM, only silibinin showed significant neuroprotective effect against Aβ$_{25-35}$ ($p < 0.01$, Fig. 1A). Further, cell viability assays showed a synergistic activity between silibinin and the two-drug combination of 5 μM curcumin and 0.5 μM vorinostat (Fig. 1B). This three-drug combination increased the ability of PC12 cells to tolerate Aβ$_{25-35}$-induced cytotoxicity. Between the control group and the curcumin-vorinostat-silibinin (CVS) and Aβ$_{25-35}$ treatment group, no differences in either CASP3 expression or TUNEL-positive cells percentage were observed (Figs. 1C and 1D), $p > 0.01$).

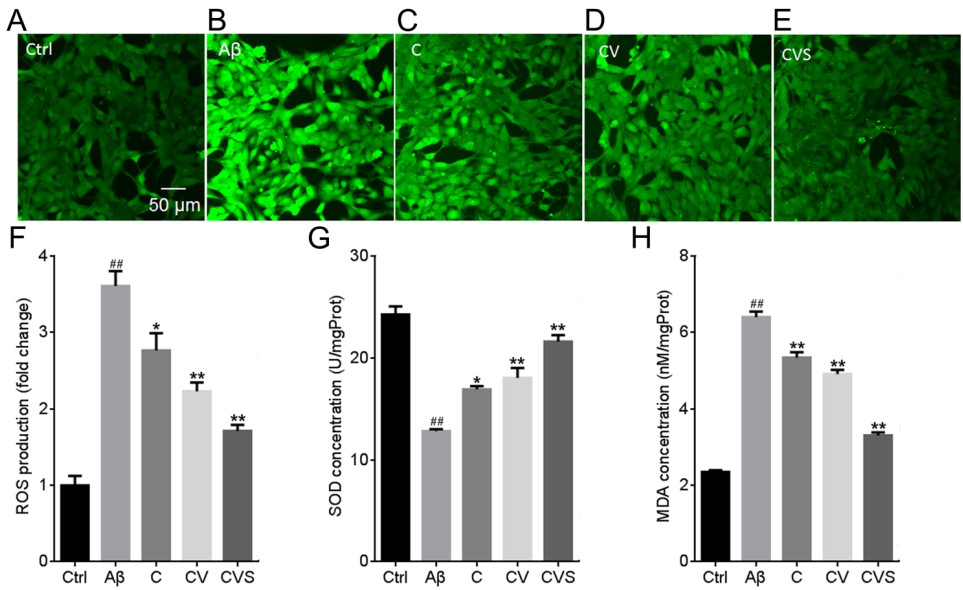

**Figure 2** **CVS pretreatment prevents Aβ$_{25-35}$-induced oxidative stress in PC12 cells.** (A–H) Effects of CVS on ROS (A–F), SOD (G), and MDA levels (H). Each experiment was completed with a minimum of five replicates. Statistical significance is presented as; ##$p < 0.001$ versus Ctrl; *$p < 0.01$, **$p < 0.001$ versus Aβ. Ctrl: control group; Aβ: Aβ$_{25-35}$ treatment group. Abbreviations: Aβ, Aβ$_{25-35}$ treatment group; C, curcumin and Aβ$_{25-35}$ treatment group; Ctrl, control group; CV, curcumin, vorinostat and Aβ$_{25-35}$ treatment group; CVS, curcumin, vorinostat, silibinin and Aβ$_{25-35}$ treatment group.

## Pretreatment with CVS significantly reduced oxidative stress following stimulation with Aβ$_{25-35}$

A significant increase in ROS production (3.6-fold relative to control) was observed in the PC12 cells treated with 20 μM Aβ$_{25-35}$ (Fig. 2A); however, in the cells pretreated with CVS for 1 h, there was no significant increase in ROS generation following Aβ$_{25-35}$ treatment ($p > 0.01$). Consistent with these results, Aβ$_{25-35}$ significantly decreased SOD concentration (1.9 fold compared to the control) and increased MDA concentration (6.4 fold compared to the control) in PC12 cells (Figs. 2B and 2C. However, in the cells pretreated with CVS, SOD and MDA concentrations were not significantly affected by Aβ$_{25-35}$ ($p > 0.01$).

## Pretreatment with CVS maintained the active status of the Akt/MDM2/p53 pathway

Treatment of PC12 cells with 20 μM Aβ$_{25-35}$ induced a significant decrease in AKT phosphorylation ($p < 0.001$, Fig. 3A), which was correlated with a significant decrease in the expression of the downstream proteins, $p$-BAD and $p$-MDM2 (Fig. 3B and 3C). However, in the cells pretreated with CVS, the Aβ$_{25-35}$-mediated reduction of these proteins was inhibited (Figs. 3A–3C). Further, pretreatment with CVS was more effective than pretreatment with curcumin alone or the two-drug combination (curcumin and vorinostat) in reducing the effects of Aβ$_{25-35}$ exposure. Similarly, pretreatment of PC12 cells with CVS reduced intracellular p53 expression, which was significantly increased by

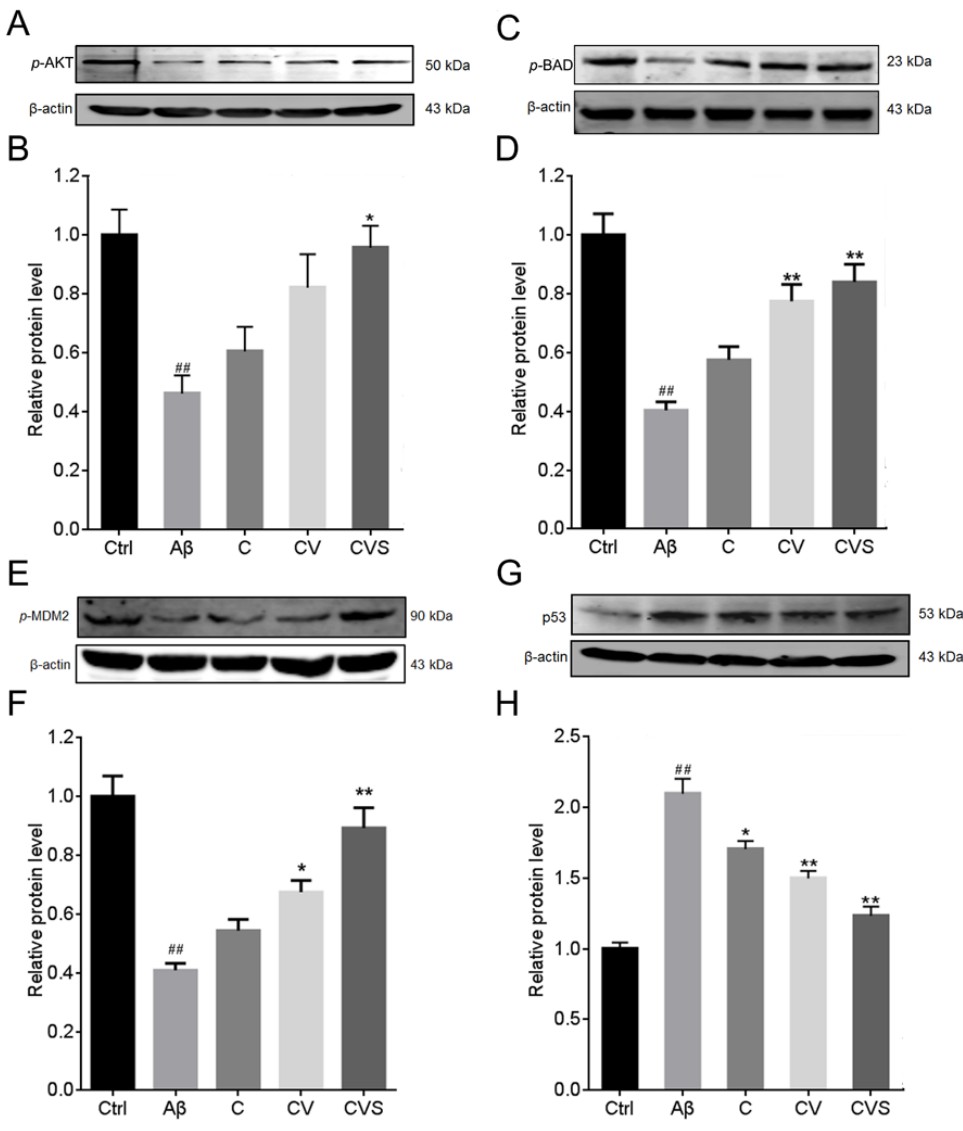

**Figure 3  Effects of CVS pretreatment on the expression of critical proteins.** (A–H) Effects of CVS on the expression of $p$-AKT (A, B), $p$-BAD (C, D), $p$-MDM2 (E, F), and p53 (G, H) in activated $A\beta_{25-35}$-treated PC12 cells. Each experiment was completed with a minimum of five replicates. Statistical significance is presented as; ##$p < 0.001$ versus Ctrl; * italic$p < 0.01$, **$p < 0.001$ versus Aβ. Abbreviations: Aβ, $A\beta_{25-35}$ treatment group; AKT, AKT serine/threonine kinase; BAD, BCL2 associated agonist of cell death; C, curcumin and $A\beta_{25-35}$ treatment group; Ctrl, control group; CV, curcumin, vorinostat and $A\beta_{25-35}$ treatment group; CVS, curcumin, vorinostat, silibinin and $A\beta_{25-35}$ treatment group; MDM2, MDM2 proto-oncogene 2; p53, tumor protein p53.

20 μM $A\beta_{25-35}$ (Fig. 3D). This was supported by a decrease in the increased transcriptional activity of p53 mediated by $A\beta_{25-35}$ in the cells pretreated with CVS, as shown by immunofluorescent microscopy results (Fig. 4).

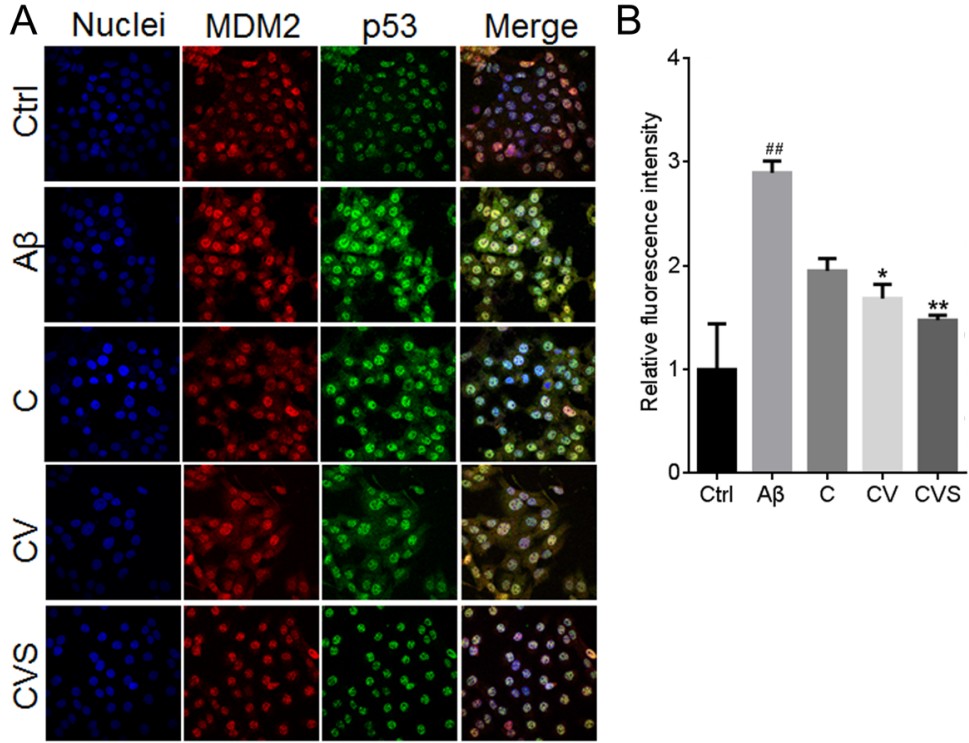

**Figure 4    CVS reduced the intranuclear content of p53 in A$\beta_{25-35}$-treated PC12 cells.** (A) Effects of CVS on the intranuclear content of p53. (B) Results of immunofluorescence staining assay. Each experiment was completed with a minimum of five replicates. Statistical significance is presented as; ##$p < 0.001$ versus Ctrl; *$p < 0.01$, **$p < 0.001$ versus A$\beta$. Abbreviations: A$\beta$, A$\beta_{25-35}$ treatment group; C, curcumin and A$\beta_{25-35}$ treatment group; Ctrl, control group; CV, curcumin, vorinostat and A$\beta_{25-35}$ treatment group; CVS, curcumin, vorinostat, silibinin and A$\beta_{25-35}$ treatment group.

## DISCUSSION

In the United States, approximately one in seven people aged $\geq 65$ years have AD, and this figure jumps to approximately 50% in those aged over 85 years (*Alzheimer's Association, 2016*). As the global population ages, these numbers continue to grow, making prevention and treatment of AD one of the most important healthcare issues of this century (*Goedert & Spillantini, 2006*). However, this is complicated by a severe lack of therapeutic options for AD.

It has been approximately 25 years since the amyloid hypothesis of AD was proposed; however, recent studies have shown that this is only a part of the story (*Hardy & Higgins, 1992*; *Armstrong, 2013*). A$\beta$ aggregation should be considered a reaction to, rather than a cause of the pathological progression of AD. The real impetus appears to be sedentary, overindulgent lifestyle causing chronic stress on the brain, which in turn accelerates brain aging (*Caruso et al., 2018*; *Mattson & Arumugam, 2018*). The clinical failure of treatment strategies involving scavenging of A$\beta$ from the brain partially supports this hypothesis regarding the pathological progression of AD (*Citron, 2010*). Further, clinical investigations suggest that approximately one in four patients with AD are not diagnosed according to
the discriminant threshold levels of Aβ plaques and Tau tangles, yet these patients still experience severe loss of hippocampal pyramidal neurons (*Mattson, 2015*), suggesting that this pathology may not be the exclusive result of advanced AD. Thus, the Aβ-scavenging strategy is too arbitrary and focused, and drug target research from a new perspective is necessary for developing effective AD drugs.

One of the most promising approaches is the upregulation of p53, which has been found to be crucial to AD development (*Jazvinšćak Jembrek et al., 2018*). Conveniently, p53 is encoded by *tp53*, one of the most thoroughly investigated genes in the human genome (*Dolgin, 2017*). Therefore, there are numerous potential drug candidates already available to modulate the p53 pathway, and they may be of use as targeted AD therapeutics. Recently, *Jazvinšćak Jembrek et al. (2018)* suggested that natural compounds may be a source of suitable drug candidates because many natural compounds are known to regulate this pathway. In our study, nine natural compounds were screened for their ability to inhibit the cytotoxic effects of excessive Aβ deposit in PC12 cells. Accordingly, 1 μM silibinin showed the greatest activity in the drug screening.

Subsequent experiments revealed that the addition of silibinin to our previously established drug combination (curcumin and vorinostat) enhanced the neuroprotective effects of this combination by activating the AKT/MDM2/p53 axis. By reconstructing the data of hippocampus gene expression in patients with AD (*Blalock et al., 2004*), we established an interaction network of proteins encoded by dysregulated genes in AD (*Meng et al., 2014*). This network clearly visualized the network hub identity of downregulated AKT. Consistently, inhibited AKT was validated to induce neuronal apoptosis (*Vázquez de la Torre et al., 2013*). A study by *Limesand, Schwertfeger & Anderson (2006)* showed the functional relationship between AKT and p53, which plays a crucial role in AD pathology (*Jazvinšćak Jembrek et al., 2018*). The authors also provided evidence of MDM2 as a critical information transmitter in the activation of AKT and suppression of p53-induced cell apoptosis.

The potent activity of this three-drug combination at low concentrations suggested synergistic drug interactions, which is consistent with the results of our previous study (*Meng et al., 2014*). Compared to monotherapies, synergistic drug combinations possess many inherent advantages, including lower doses, multi-target regulation, and reduced risk of drug resistance development (*Zimmermann, Lehár & Keith, 2007*; *Lehár et al., 2009*; *Jia et al., 2009*). More importantly, the three drugs investigated in this study are all commercially available. Thus, the CVS treatment is economically feasible for late-stage drug development.

Curcumin is a naturally found flavone chemical, and studies have shown that it is a promising anti-AD drug (*Venigalla, Gyengesi & Münch, 2015*). However, its poor bioavailability in humans limits its further clinical application (*Chin et al., 2013*). Multiple strategies have been assessed to solve this issue of low bioavailability. For example, formulation of curcumin-loaded nanoparticles increased curcumin bioavailability, but it was associated with higher costs (*Tiwari et al., 2014*). Other studies have indicated that a curcuminoid mixture (instead of curcumin alone) has great efficacy in the potential treatment of AD (*Ahmed & Gilani, 2014*), suggesting that combination therapies may

be an effective strategy for using curcumin as a treatment for AD. Vorinostat, or suberoylanilidehydroxamic acid, is a histone deacetylase inhibitor approved by the FDA as a treatment for cutaneous T cell lymphoma. Vorinostat may also have potential value as a treatment for AD through its effects on the CREB-binding protein (CBP)/E1A-binding protein p300 (EP300) signaling pathway (*Rouaux, Loeffler & Boutillier, 2004*). In a previous study, we showed synergistic properties between vorinostat and curcumin in protecting PC12 cells against Aβ toxicity; however, exposure to high concentrations of vorinostat was found to be cytotoxic (*Meng et al., 2014*). Given this potential cytotoxicity, we reduced its concentration from 1 μM to 0.5 μM. Recently, several studies have shown the neuroprotective effects of silibinin in a model of Aβ-treated rats (*Song et al., 2017*; *Song et al., 2018*). Consistent with these findings, our results showed that silibinin increased the effects of the two-drug combination in protecting neural cells from Aβ toxicity.

In our study, the experiment was designed such that PC12 cells were pretreated with the drug combination prior to treatment with Aβ aggregation. Based on the clinical features of AD, three developmental stages can be defined: pre-clinical AD, prodromal AD, and AD-type dementia (*De-Paula et al., 2012*). The former two phases are the pre-symptomatic phase of AD, whereas the latter is the symptomatic phase. The pre-symptomatic phases of AD last longer than its symptomatic phase (*De-Paula et al., 2012*), and they are associated with lower economic costs (*Jia et al., 2018*). Therefore, preventing dementia via treatment of early-stage AD is more economically feasible than treating AD-type dementia (*Goedert & Spillantini, 2006*).

## CONCLUSIONS

Through a cell viability screening of nine natural compounds, we successfully identified CVS, a combination of three low-concentration drugs, which had potential as an AD therapeutic. Our results showed the strong neuroprotective ability of CVS against Aβ toxicity *in vitro,* and that pretreatment with CVS increased the tolerance of nerve cells to Aβ toxicity. We also showed that simulated Aβ aggregation led to inactivation of the AKT/MDM2/p53 pathway, which has a critical role in AD progression. However, pretreatment with CVS maintained the active state of the pathway and ensured low transcriptional activity of p53. As a direct result of the protection provided by CVS, no significant cell apoptosis or oxidative stress occurred when PC12 cells were exposed to A*β* aggregation. However, this verification of the effectiveness of CVS in vitro is only the first step. Further studies in animal models are necessary to evaluate the therapeutic value of CVS for AD. In conclusion, our findings suggested that CVS is a promising prophylaxis for AD treatment.

## ACKNOWLEDGEMENTS

We thank Dr. Songbin Fu for reviewing the manuscript and giving suggestions.

### Funding

This study was supported by the Heilongjiang Science Foundation (QC2016113) and the Heilongjiang postdoctoral Foundation (LRB14-315). The funders had no role in study design, data collection and analysis, decision to publish, or preparation of the manuscript.

### Competing Interests

The authors declare there are no competing interests.

### Author Contributions

- Jia Meng performed the experiments, analyzed the data, prepared figures and/or tables.
- Yan Li analyzed the data.
- Mingming Zhang, Lin Zhou, Qiujun Wang and Lin Lin performed the experiments.
- Wenjing Li conceived and designed the experiments, performed the experiments.
- Lihong Jiang contributed reagents/materials/analysis tools, authored or reviewed drafts of the paper.
- Wenliang Zhu authored or reviewed drafts of the paper, approved the final draft.

### Data Availability

The raw data for all the figures are available in a Supplemental File.

### Supplemental Information

Supplemental information for this article can be found online at http://dx.doi.org/10.7717/peerj.6716#supplemental-information.

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
