# Peer review of "A combination of curcumin, vorinostat and silibinin reverses Aβ-induced nerve cell toxicity via activation of AKT-MDM2-p53 pathway"

_PeerJ, doi:10.7717/peerj.6716_

## Round 0.1 · original submission · Major Revisions

Based on the advice received, I have decided that your manuscript could be reconsidered for publication should you be prepared to incorporate major revisions. However, we are not prepared to accept your manuscript in its present form.

Reviewer 1 ·

Basic reporting

1) The authors use a clear and adequate English language throughout the manuscript;
2) The intro & background presented are well referenced and relevant to the context;
3) The manuscript structure follows the PeerJ standards;
4) The figures are relevant, well labelled and described. However, I may have some suggestions to improve their quality:
a) I suggest authors remove the frames on the top and right of figures… it will make a better presentation to the audience;
b) The most important issue lies in Figure 3C. The actin band does not appear to be homogeneous, which may indicate errors in sample concentration. This is an essential condition to calculate the real value of proteins. Therefore, errors in protein content can display false-positive results. This important issue should be fixed, as the results are not reliable.

Experimental design

1) The authors present original primary research related to a well defined, relevant and meaningful question, which fills an identified knowledge gap.
2) The investigation was performed according to a high technical and ethical standard, describing methods with enough detail and information to replicate.

Validity of the findings

1) The authors present meaningful findings, statistically sound and controlled. However, the description written in the manuscript’s “Discussion” section is not informative to properly interpret the obtained data. The authors repeat their results throughout the discussion, instead of providing the proper mechanisms by which the CVS combination activates AKT-MDM2-p53 pathway.
2) The conclusions are well stated, linked to the original research question and limited to supporting results. Yet, the authors state that the CVS combination has potential as prophylaxis for AD treatment, but they only performed in vitro assays, which can seem as a speculative approach. Maybe authors could consider stating it as a first step, which leads to further investigations using animal models (as mentioned in line 280-281 of the manuscript).

Additional comments

The manuscript presents a promising work which provides interesting approaches to the search for new therapeutic options in the different phases of AD. The authors suggest that the curcumin, vorinostat and silibinin combination protects PC12 cells from Aβ25-35-induced toxicity through modulation of AKT/MDM2/p53 pathway. However, some accuracies are necessary to make clear some issues.

Annotated reviews are not available for download in order to protect the identity of reviewers who chose to remain anonymous.

·

Basic reporting

The article is very well written and the English is clear.

However, the Introduction needs more detail. There was no discussion of why beta-amyloid 25-35 was selected to stress the cells, nor did it clearly describe the author’s previous work with this compound in the cultured cell model. Adding this detail would enhance the paper and it needs to be included in order for the paper to be published.

Experimental design

The methods are well described. The inclusion of proper control groups is a concern, as noted below under validity.

Validity of the findings

The findings appear valid, but the results and figures need to be more clearly presented.
1. Figure 1 needs additional explanation. In the figure, a control and then A is shown, and then the effects of 9 potentially protective compounds. Is this in the presence of the compound plus A or the compound alone. In part B of the figure, there is the same issue. Also, what is the effect of curcumin, voinostat and silibinin alone? Those are the proper controls, which are not shown.
2. The Results section has similar issues, and needs to be more clear about what was tested and what were the results with the compounds alone.
3. The same issues apply to Figures 2, 3 and 4. While the reader can figure out what the authors mean, it should be clear in the Results section and in the figure legends.

Additional comments

This is a very interesting paper, but it would benefit from increased clarity of the experimental design and results. It discusses a novel approach to Alzheimer's disease that should be pursued.

Reviewer 3 ·

Basic reporting

- There are some grammatical and typographical mistakes along the manuscript which should be corrected. In many instances throughout the manuscript, it was difficult to understand what the authors were trying to convey.
- The abstract should be rewritten. The abstract must state the hypothesis and objective for the research and contain the experimental procedure briefly.
- The introduction needs more detail. I suggest that you improve the description at lines 39-50 to provide more justification for your study bringing current data on the number of people with AD worldwide and not just in some countries.
- Some references are not formatted correctly. Authors should review this section.

Experimental design

- The authors do not provide references in the methods section. Authors should review and revise this section.
- It is not clear in the methodology because the authors used the combination with CVS as pretreatment. They should explain why choosing the pretreatment protocol. Why did not they use the combination of CVS as treatment after stimulation by Aβ25–35.

Validity of the findings

- The work presented by the authors is not mechanistic and it is not novel. Why is the study important?
- No differences were observed between the control group and the curcumin-vorinostat-silibinin (CVS) group with regard to either CASP3 expression or the percentage of TUNEL positive cells. How do the authors explain this results?
- Authors should explain the applicability of the results found.
- The authors investigated whether the addition of silibinin could act synergistically with curcumin and vorinostat to alleviate amyloid beta (Aβ)-induced nerve cell toxicity. From the results obtained, it was evident that the combination showed positive effects on the parameters analyzed in the study, however the discussion is very poor and needs to be rewritten in more detail. It is necessary that the authors discuss in depth the results of the study and propose a possible mechanism to explain the benefits of curcumin, vorinostat and silibinin.

---

## Round 0.2 · accepted · Accept

Based on the advice received and my own evaluation, I have decided that your manuscript could be accepted for publication in PeerJ in its present form.

# ·

Basic reporting

The authors have revised the manuscript and answered the reviewers criticisms.

Experimental design

The authors have answered the reviewers concerns and revised the figures to reflect these.

Validity of the findings

No comment.

Additional comments

The authors have made a strong attempt to reply to the reviewers concerns.

Reviewer 3 ·

Basic reporting

no comment

Experimental design

no comment

Validity of the findings

no comment

Additional comments

The authors have replied in a satisfactory manner. According to this reviewer, the article can be accepted for publication.